# An Experimentally Induced Mutation in the UBA Domain of p62 Changes the Sensitivity of Cisplatin by Up-Regulating HK2 Localisation on the Mitochondria and Increasing Mitophagy in A2780 Ovarian Cancer Cells

**DOI:** 10.3390/ijms22083983

**Published:** 2021-04-13

**Authors:** Sihang Yu, Xiaoyu Yan, Rui Tian, Long Xu, Yuanxin Zhao, Liankun Sun, Jing Su

**Affiliations:** Department of Pathophysiology, College of Basic Medical Sciences, Jilin University, Changchun 130000, China; 18686448815@163.com (S.Y.); yanxiaoyu0713@163.com (X.Y.); tianrui18@mails.jlu.edu.cn (R.T.); xulong199609@outlook.com (L.X.); hepta@foxmail.com (Y.Z.)

**Keywords:** mitophagy, UBA, p62, cisplatin, apoptosis, ovarian cancer

## Abstract

The study of cisplatin sensitivity is the key to the development of ovarian cancer treatment strategies. Mitochondria are one of the main targets of cisplatin, its self-clearing ability plays an important role in determining the fate of ovarian cancer cells. First, we proved that the sensitivity of ovarian cancer cells to cisplatin depends on mitophagy, and p62 acts as a broad autophagy receptor to regulate this process. However, p62′s regulation of mitophagy does not depend on its location on the mitochondria. Our research shows that the mutation of the UBA domain of p62 increases the localisation of HK2 on the mitochondria, thereby increasing the phosphorylated ubiquitin form of parkin, then stabilising the process of mitophagy and ultimately cell survival. Collectively, our results showed that a mutation in the UBA domain of p62 regulates the level of apoptosis stimulated by cisplatin in ovarian cancer.

## 1. Introduction

Ovarian cancer is highly prevalent in developed countries and is associated with a high mortality rate. Cisplatin is a common drug for treating ovarian cancer. However, sensitivity of ovarian cancer cells to cisplatin varies. In fact, ovarian cancer is mainly divided into three categories according to the origin: epithelial origin, germ cell origin and specific stromal cell tumors. Among them, the vast majority of ovarian cancers are of epithelial origin (EOCs) [1,2]. The A2780 cell line selected in this study is an untreated human ovarian tumor of epithelial origin. It is widely used in cytotoxicity and cancer research, especially in the study of cisplatin sensitivity to ovarian cancer. A2780/DDP is a cisplatin resistant strain of the A2780 cell line. Therefore, the A2780/DDP cell line is often used in the study of platinum-resistant ovarian cancer models [3,4].

Mitochondria, being the central for energy and signal transduction in cells, serves as one of the primary targets for chemotherapeutic drugs. Mitochondria are important targets for platinum-based chemotherapeutics to act on the cancer cells and participate in the process of apoptosis stimulation and execution [5,6]. However, the accumulation of reactive oxygen species (ROS) and reduction of membrane potential caused by this process promote mitophagy [7,8].

Mitophagy is a form of selective autophagy, which responds to certain specific stress (such as drug stimulation, hypoxia and nutrient deprivation, etc.) in an acute response to clear the damaged or excess mitochondria in the mitochondrial pool [9]. According to the characteristics of mitophagy, it is divided into four stages. (A) After the mitochondria is damaged, the permeability changes, depolarisation occurs (membrane potential decreases), and mitochondrial autophagy-related proteins are activated. (B) Early autophagosomes wrap damaged mitochondria with the participation of LC3 to form mitophagosomes (MP). (C) Mitophagosomes use acetylated tubulin as a scaffold to combine with lysosomes to form mitolysosomes (ML). (D) Mitochondria are finally degraded in lysosomes [10]. The important proteins involved in this process include phosphatase and tensin homolog-induced kinase 1 (PINK1)/parkin, NIX/BNIP3, and FUN 14 domain-containing protein 1 (FUNDC1) [11]. PINK1/parkin is involved in mitophagy stimulated by cisplatin [7,12]. The cisplatin-induced PINK1/parkin-mediated mitophagy is considered protective [13,14,15]. The main process is the decrease of mitochondrial membrane potential in damaged mitochondria [6]. PINK1 fails to shear the mitochondrial inner membrane and is fixed on the mitochondrial outer membrane. Parkin is recruited through autophosphorylation [16], And extensively ubiquitinates the mitochondrial outer membrane proteins, such as voltage-dependent anion channel 1 (VDAC1), mitofusin 1 (MFN1), and MFN2 [17]; some of these proteins provide a docking site for the LC3 interacting, including proteins p62/SQSTM1 and NBR-1, and upon successful docking this results in fusion of the mitophagosomes with lysosomes forming mitophagosomes–lysosomes that achieve degradation of the damaged mitochondria [18,19,20].

The subcellular localisation of some proteins such as parkin, hexokinase 2 (HK2), and the B-cell lymphoma 2 (BCL2) family of proteins within the mitochondria affects its “life activities” [21,22,23,24]. During mitophagy, a combination of phosphorylated ubiquitin (p-Ub) and parkin is believed to activate parkin and stabilise its localisation in the mitochondria [25,26]. HK2, one of the key enzymes in the glycolysis process, affects the “feeding mode” of cells, and it is regulated by glycogen synthase kinase-3 beta (GSK3β)/protein kinase B (AKT). HK2 binds to VDAC1, both of which affect the opening of mitochondrial permeability transition pore (mPTP) and regulate the release of cytochrome C (Cyto C) to determine the cell fate [22,27,28,29]. Autophagy is the main degradation pathway of HK2 in liver cancer compared to the degradation by the proteasome pathway. The primary autophagy receptor for HK2 degradation is p62 [30]. Under normal circumstances, HKII is located on both cytoplasm and mitochondria. Its role is largely involved in glycolysis to provide energy for cells. However, HKII located on mitochondria will further promote the process of glycolysis. The reason is that ATP produced in the mitochondria will be released into the cytoplasm through mPTP, then the combination of HKII and VDAC1 which constitutes mPTP will get a nearby energy supply, thereby promoting the occurrence of glycolysis. Therefore, the increase of mitochondrial localisation of HKII is often found in cancer cells and even drug-resistant cancer cells. Mitochondrial localisation of HKII has become the main target of cisplatin resistance research [28,31].

Cisplatin is a multi-target anti-tumor drug, and mitochondria are also one of its main targets. And as mentioned above, the mitochondrial localisation of HKII is regulated by GSK3β/AKT. In the study of nasopharyngeal carcinoma, it was found that Skp2 increased the expression of HKII. At the same time, it could also increase AKT-targeted mitochondrial ubiquitination and phosphorylation. Finally, the localisation of HKII on the mitochondria was increased. This is an important molecular mechanism of cisplatin resistance in nasopharyngeal carcinoma [32], and due to the differences in tumor types, individual differences in patients and even the heterogeneity of cancer cells in the tumor, the mitochondrial localisation changes of HKII under cisplatin stimulation are not clear, and this is also worth studying.

The multiple domains of p62, referred to as the signal hubs, bind a variety of key molecules involved in cell life activities [33,34,35]. However, the role of p62 as an autophagy receptor in mitophagy is no different from that in macroautophagy. In mitophagy, p62 binds to the ubiquitinated proteins through its ubiquitin-associated (UBA) domain and anchors to LC3 in the autophagosome membrane through its LC3II-interacting region (LIR) domain [36]. PINK1/parkin-mediated mitophagy depends on p62/SQSTM1 [37]. So, this study aimed to explore the molecular mechanism that affects the sensitivity of ovarian cancer to cisplatin through the regulation of the UBA domain of p62 on mitophagy.

Here, we verified the involvement of p62 in the cisplatin-induced mitophagy in ovarian cancer cells. and found that when the UBA domain is knocked out, the mitochondrial localisation of parkin and HK2 increased, and then found that ovarian cancer cell mitophagy increased, which decreases the drug sensitivity to cisplatin. It suggests that the location of parkin and HK2 in cells will affect the physiological process of mitochondria, that is, mitophagy, and then affect the prognosis of ovarian cancer.

## 2. Results

### 2.1. p62 Is Involved in the Mitochondrial Clearance of Ovarian Cancer Cells Stimulated by Cisplatin

We used the 3-(4,5-dimethylthiazol-2-yl)-2,5-diphenyltetrazolium bromide (MTT) assay to test the cisplatin sensitivity of A2780 and A2780/DDP cell lines. While the A2780/DDP cells had high cell viability with a half maximal inhibitory concentration (IC50) of approximately 22 μg/mL, the IC50 for A2780 cell line was approximately 5.5 μg/mL (Figure 1a). We chose 5 μg/mL and 20 μg/mL as the final concentrations of cisplatin for A2780 and A2780/DDP cell lines, respectively. The self-clearance ability of mitochondria is considered as one of the various reasons for the differential cisplatin sensitivity. Hence, we used MTT to detect the effects of cyclosporin A (CsA) on the viability of A2780/DDP cells. The results revealed low-dose CsA to have no obvious inhibitory effect on the A2780/DDP cells (Figure 1b). However, when we added mitophagy inhibitors CsA (10 μM) and cisplatin to A2780/DDP cells and detected the cell viability using MTT assay, IC50 of cisplatin was determined to be 7.1 μg/mL (Figure 1c).

Next, we used CompuSyn software to analyse the cell index (CI) data [38,39]. When the dose of cisplatin was ≥8 μg/mL, the CI value was less than 1, and the two drugs exerted a combined effect (Figure 1d). It can be seen from the drug resistance index value that at the level of inhibition observed in the experiment, the combined medication can reduce the dosage of cisplatin by 1.09191–3.27583 times (Figure 1e).

Mitochondria are the frequent targets of stress resulting in the opening of nonselective high conductance mPTP that causes mitochondrial depolarisation, a condition marking the initiation of mitophagy. When the opening of mPTP is inhibited, CsA reverses the effect of reduced mitochondrial membrane potential caused by stimulating factors, thereby inhibiting mitophagy [31,32]. Carbonyl cyanide p-trifluoromethoxyphenylhydrazone (FCCP) can increase the permeability of the mitochondrial inner membrane to hydrogen ions (H+), eliminate H+ gradients, and decrease the mitochondrial membrane potential. Therefore, the occurrence or inhibition of mitochondrial depolarisation is the reason for choosing cisplatin or CsA [33].

We speculate mitophagy to be the reason for the low cisplatin sensitivity of A2780/DDP cells. We tested the co-localisation of LC3II within the mitochondria in the presence of mitophagy agonists FCCP as a positive control and cisplatin and CsA as the negative controls. A2780/DDP cells had more co-localised LC3II and exhibited greater mitochondrial fluorescence when untreated than A2780 cells (Figure 1f (①)), thus, suggesting a higher level of mitophagy in A2780/DDP cells. The co-localisation of LC3II and mitochondrial fluorescence in A2780/DDP cells significantly increased under short-term cisplatin stimulation (Figure 1f (②)). Cisplatin and FCCP groups showed similar phenomena (Figure 1f (③))

As a signal hub and an important autophagy receptor, p62 may be involved in the regulation of mitophagy in A2780/DDP cells stimulated by cisplatin. We found increased expression of the proteins parkin, PINK1, and p62 in A2780/DDP cells (Figure 2a). We then added cisplatin in A2780/DDP cells, using FCCP as the positive control. Both the groups showed that the fluorescence co-localisation site of p62 and mitochondria increased, and p62 was recruited to the mitochondria (Figure 2b). To further study the effect of p62 on mitophagy involved in the parkin/PINK1 axis, we knocked down the expression of p62 encoding gene in A2780/DDP cells and found that the recruitment of parkin and LC3II to the mitochondria under cisplatin stimulation was reduced (Figure 2c,d).

### 2.2. Regulation of Mitophagy by p62 Does Not Depend on the Location of p62 within the Mitochondria

To re-confirm the role of p62 in mitophagy, we overexpressed wild-type and UBA domain truncated mutant p62 (ΔUBA) (Figure 3a). The above two plasmids were transfected into A2780 cells, and the cytoplasmic and mitochondrial proteins were separated. Expression of p62 was reduced in the mitochondrial lysate of the ΔUBA group (Figure 3b). We also detected the co-localisation of p62 and mitochondrial outer membrane marker protein VDAC1 during mitophagy by immunoprecipitation (IP) and found that the combination of p62 and VDAC1 in the ΔUBA group was reduced (Figure 3c). At the same time, we also used immunofluorescence to detect the fluorescence co-localisation of p62 and mitochondria when the UBA truncation mutation was stimulated by cisplatin. The results revealed decreased mitochondrial localisation of p62 after UBA truncation (Figure 3d). Since the tertiary structure of p62 may have changed after the truncation of the UBA domain, we constructed a mutant form of p62 with the mutation at position 417 of UBA. After short-term cisplatin stimulation, the mitochondria and cytoplasmic proteins were separated. However, still no improvement in the mitochondrial localisation of p62 was reported in the UBA417 mutant group (Figure 3e).

We speculate that the mitochondrial localisation of p62 may depend on its UBA domain, and this may affect its regulation of mitophagy. Therefore, this study explored from the three perspectives of mitophagy in the early, late and end-stage whether it still has a regulatory effect on mitophagy when the p62 domain is deleted.

This research used cisplatin to stimulate the ovarian cancer cells for a short period and then stained them with JC-1. According to the principle of JC-1 staining, when the membrane potential is normal, JC-1 enters the mitochondria through the polarity of the mitochondrial membrane and forms a red fluorescent polymer owing to the increased concentration. When the mitochondrial transmembrane potential is depolarised, JC-1 is released from the mitochondria, the concentration is reduced, and it reverses to its monomer form emitting green fluorescence. Therefore, changes in the mitochondrial membrane potential can be detected qualitatively (shift in the cell population) and quantitatively (fluorescence intensity of the cell population) by detecting green and red fluorescence.

Next, we studied the effect of the UBA domain of p62 on mitophagy. When wild-type p62 and ΔUBA were transfected into A2780 cells after stimulation with cisplatin, JC-1 staining results showed an obvious decrease in the mitochondrial membrane potential of the ΔUBA group (Figure 4a). The lentivirus mt-keima-COX8 was used to transduce the A2780 cell line. It is worth noting that keima is a pH-sensitive fluorescent protein. When it enters the lysosome cavity from the neutral environment of the mitochondria, fluorescence emission changes from green to red. Flow cytometry and fluorescence resulted in the ΔUBA group revealing an increase in the ratio of red to green light signal under cisplatin stimulation. The green light means that keima is still in the neutral environment of mitochondria. The ratio of red light to green light represents the flow of mitophagy. Therefore, we used the ratio of the upper left area (cells where mitophagy is highly activated) and the lower right area (cells with almost no mitophagy) to quantify the level of mitophagy. Mitophagy in ovarian cancer cells also increased significantly (Figure 4b,c). To further evaluate the quality of mitochondria in ovarian cancer cells after knocking out the UBA domain, real-time polymerase chain reaction was performed. A significant decrease in the copy number of mitochondrial DNA (mtDNA) was observed in the ΔUBA group (Figure 4d).

### 2.3. Recruitment of HK2 to the Mitochondria Is the Key to the UBA Domain-Mediated Mitophagy Regulation in Ovarian Cancer

Following the above experiments, we found an interesting phenomenon. Mitophagy was blocked upon knocking down the gene encoding full-length p62. However, when only the UBA domain was deleted, the level of mitophagy increased significantly. The contradictory results could be explained by the diversity and versatility of the p62 domain. Furthermore, it was apparent that the different domains of p62 play different roles in the process of mitophagy.

To further explore the effect of the UBA domain truncation of p62 on PINK1/parkin-mediated mitophagy, we tested the combination of parkin and p-Ub by IP and found their increased expression in the ΔUBA group stimulated with cisplatin (Figure 5a). We then isolated the mitochondrial proteins to explore which kinase affected parkin-related phosphorylation and stabilised mitophagy. No significant change in PINK1 was noted. However, recruitment of HK2 to mitochondria increased (Figure 5b). The results of co-IP with VDAC1 and fluorescence co-localisation with mitochondria also showed an increased localisation of HK2 in mitochondria (Figure 5c,d). To determine the relationship between p62 and HK2, we knocked down the expression of p62 encoding gene in A2780/DDP cells and found an increased expression of HK2 (Figure 5e). After A2780 cells were transfected with wild-type p62 and ΔUBA, expression of HK2 in the ΔUBA group increased (Figure 5f) under stimulation with cisplatin. When the UBA domain was deleted, binding of p62 to HK2 reduced (Figure 5g).

Therefore, p62 serves as an important autophagy receptor of HK2 in ovarian cancer cells and participates in the degradation of HK2. It may change its mitochondrial localisation by changing the overall abundance of HK2 in the cell. HK2 increases the phosphorylation of parkin, which eventually stabilises and increases the level of mitophagy.

### 2.4. A Mutation in the UBA Domain of p62 Causes Balanced Regulation of Mitophagy by Altering the Cisplatin Sensitivity of Ovarian Cancer Cells

From the previous experiments, it can be concluded that when the UBA domain of p62 is truncated, HK2 promotes PINK1/parkin-mediated mitophagy. Mitophagy has been proven to be an important factor for regulating tumour cell platinum drug tolerance. We also questioned whether the UBA domain could regulate cisplatin sensitivity of ovarian cancer cells by controlling the mitochondrial turnover.

Hence, we used MTT assay to test the viability of the cells to cisplatin treatment when the UBA domain of p62 was knocked out. The ΔUBA group exhibited a higher survival rate than the wild-type p62 group at a concentration of 10–20 μg/mL of cisplatin (Figure 6a). It is speculated that the ΔUBA group may have a lower sensitivity to cisplatin. So, in order to further explore the differences in the cell proliferation ability of these groups under cisplatin stimulation, we then observed the cells’ response to cisplatin using a real-time cell analysis platform. After culturing for 24 h, cisplatin was added to the transfection group, and it was found that the CI was higher in the ΔUBA group (Figure 6b). It shows that the proliferation ability of ΔUBA group is the least susceptible to interference from cisplatin in these treatment groups.

Based on the previous results, we found that the mitochondrial localisation of HKII increased and the binding to VDAC1 increased. Therefore, it is inferred that this process improves the stability of the mitochondrial membrane and reduces the apoptosis of the mitochondrial pathway. We thus evaluated the level of apoptosis.

Hoechst staining revealed that the nuclear fragmentation induced by cisplatin treatment was significantly reduced in the cells overexpressing ΔUBA compared to that in the wild-type p62 group (Figure 6c). We used annexin V/propidium iodide staining and Western blotting to detect apoptosis in each group after adding cisplatin and observed a proportion of apoptotic cells in the ΔUBA group (Figure 6d). At the same time, the ratio of Bcl2 to Bax increased and the expression of cleaved caspase 3 (c-caspase 3) and cleaved poly (ADP-ribose) polymerase (c-PARP) decreased (Figure 6e). It can be found that ΔUBA group improves the tolerance to cisplatin stimulation by reducing the apoptosis of the mitochondrial pathway.

## 3. Discussion

In this study, we explored the role of p62 in mitophagy and its relationship with the cisplatin sensitivity of A2780 cells. Knockout of the UBA domain of p62 resulted in accumulation of HK2 and its recruitment to the mitochondria, which thereby increased parkin-mediated mitophagy in A2780 cells. The ability of A2780 cells to eliminate damaged mitochondria was enhanced resulting in altered cell fate (Figure 7).

Apoptosis in the mitochondrial pathway is a precisely regulated form of cell death that is used as a tumor suppressor mechanism by targeting cancer cells. Therefore, mitophagy caused by chemotherapy drugs is often considered to be a self-protection mechanism of cancer cells and one of the reasons for chemotherapy resistance [40]. In this study, the relationship between mitophagy and apoptosis is explored using a variety of methods that have been used earlier to detect the level of mitophagy [41].

Being a multifunctional scaffold protein, p62 serves as a signalling hub by interacting with the core molecules involved in a variety of important cell life activities [42,43], An increasing number of studies have proved that the integrated analysis of the expression of p62 and other key proteins in signalling pathways can be used as a biomarker for cancer prognosis [44,45], especially including ovarian cancer [2,46]. Studies have found that the ZZ domain of p62 in ovarian cancer regulates the activation of the nuclear factor kappa-light-chain-enhancer of activated B cells (NF-kB) pathway mediated by receptor-interacting protein 1 (RIP1) and affects the proliferation and apoptosis sensitivity of ovarian cancer cells [47]. In addition, p62 and caspase 8 are responsible for the progression of ovarian cancer [48].

It has been also reported that the KIR domain of p62 can bind to kelch-like ECH-associated protein 1 (KEAP1) and increase the level of mitophagy in cells by up-regulating the ubiquitination level of mitochondrial surface proteins. When the KIR domain was inhibited, the level of mitophagy decreased [40]. However, the UBA domain of p62 inhibited this process, indicating that the regulation mechanism of p62 for mitochondrial autophagy is complicated. Our results also showed that when the UBA domain is knocked out, the anchoring of p62 on the mitochondria is significantly reduced, thereby increasing the level of mitophagy. We speculate that regulation of mitophagy by p62 does not depend on its location on the mitochondria.

However, few studies have revealed the involvement of the p62 UBA domain in the regulation of the subcellular localisation of certain proteins, such as parkin and HK2. The altered localisation of these proteins may influence cell survival or death. In the course of exploring the sensitivity of ovarian cancer cells to cisplatin, this study first proposed the role of UBA domain of p62 in regulating the recruitment of HK2 to mitochondria. Protein is the main bearer of cell life activities, but protein is not concentrated in a specific location within the cytoplasm. Therefore, insight into the localisation of mitochondrial proteins in cells is of great importance in understanding the physiological processes of mitochondria and diseases related to mitochondrial [49,50]. In the past 30 years, there have been many computational models and methods for predicting the subcellular localisation of proteins. Also multiple methods can be employed to extract characteristic information of protein sequences that helps in predicting the subcellular localisation of proteins [51,52].

As one of the key enzymes of glycolysis, HK2 binds to the outer mitochondrial membrane protein VDAC1 and thereby directs ATP produced in the mitochondria preferentially enter HK2 for glycolysis [27,53]. HK2 in combination with VDAC1 affects cell apoptosis by controlling Mptp [54]. In our experiment, the increased binding of HK2 and VDAC1 may be one of the reasons for the decreased apoptosis rate of cisplatin in the ΔUBA group in addition to the increased mitochondrial clearance efficiency. At the same time, we also speculate that the increased localisation of HK2 in mitochondria may change the metabolic pattern of the ΔUBA group, that is, increase the glycolytic flow.

In conclusion, this study involving discussion on the UBA domain of p62 regulating the mitochondrial localisation of HK2 and thus changing the cisplatin sensitivity of ovarian cancer cells can be better understood as the study of the mechanism of subcellular protein localisation for the prognosis of ovarian cancer. Also, it provides a lead for the development of mitophagy inhibitors and HK2-targeted inhibitors.

## 4. Materials and Methods

### 4.1. Cell Lines and Culture

Human ovarian carcinoma cells (A2780 and A2780/DDP cells) were purchased from American Tissue Culture Collection (Rockville, MD, USA).

A2780 and A2780/DDP ovarian cancer cells were grown in RPMI-1640 (Gibco Life Technologies, Carlsbad, CA, USA) supplemented with 10% foetal bovine serum (Invitrogen, Carlsbad, CA, USA) at 37 °C at 5% CO_2_ concentration.

The reagents used in this study include the following: cisplatin (Sigma-Aldrich, St. Louis, MO, USA), FCCP (Sigma-Aldrich), Cyclosporin A (Sigma-Aldrich), 3-(4,5-dimethylthiazol-2-yl)-2,5-di-phenyltetrazolium bromide (MTT) (Sigma-Aldrich), ViaFect™ transfection reagent (Promega, Madison, MI, USA), MitoTracker Red CMXRos (Sigma-Aldrich). Antibodies used in this study include anti-p62 (Abcam, Cambridge, MA, USA, ab109012), anti-Pakin (Proteintech, Chicago, IL, USA, 14060-1-AP), anti-PINK1 (Proteintech, Chicago, IL, 23274-1-AP), anti-HK2 (Proteintech, Chicago, IL, USA, 22029-1-AP), anti-VDAC1 (Proteintech, Chicago, IL, USA, 55259-1-AP), anti-actin (Proteintech, Chicago, IL, USA, 66009-1-Ig), anti-phospho-ubiquitin (Millipore, Burlington, MA, USA, ABS1513-I), anti-Bax (Abcam, Cambridge, MA, USA, ab32503), anti-Bcl-2 (Abcam, Cambridge, MA, USA, ab32124), anti-caspase3 (Abcam, Cambridge, MA, USA, ab32351), anti-c-caspase3 (Abcam, Cambridge, MA, USA, ab32042), anti-c-parp (Abcam, Cambridge, MA, USA, ab32064) and anti-CoxIV (Santa Cruz, CA, USA, sc-376731).

### 4.2. Plasmid and Transfection

The pcDNA3.1 vector (NC), pcDNA3.1-p62, pcDNA3.1-ΔUBA-p62, pcDNA3.1-L417V UBA-p62, si-p62 (①, ②) were purchased from Sangon Biotech (Shanghai, China). Cells were transfected using ViaFect™ transfection reagent according to the manufacturer’s instructions. Take a 6-well plate as an example, add 4 μg plasmids or 50 pmol siRNA to each well. The culture system is 2 mL per well.

### 4.3. Cytotoxicity Assays

Cells (8 × 10^3^ cells per well) were plated in 96-well plates and treated with drugs for 24 h. MTT reagents were added and cells were incubated for 4 h. Absorbance values were then measured at 570 nm using a Vmax Microplate Reader (Molecular Devices, LLC, Sunnyvale, CA, USA).

### 4.4. Western Blotting

After treatments, cells and tumors were lysed with lysis buffer (TaKaRa, Tokyo, Japan). Western blotting was carried out as previously described. Protein content was determined using Bradford reagent (Bio-Rad, Hercules, CA, USA). Protein blot signals were detected with an enhanced chemiluminescence detection kit (DW101, TransGen Biotech, Beijing, China).

### 4.5. Mitochondria Separation

Mitochondria extraction was conducted using a Mitochondria Isolation Kit (Invent Biotechnologies, Plymouth, MN, USA) according to the manufacturer’s protocol.

### 4.6. Co-Immunoprecipitation

After treatments, cells were lysed in NP40 lysis buffer. Immunoprecipitation assays were carried out as previously described using HK2, Parkin and VDAC antibodies, and protein G agarose (Beyotime Biotechnology, Shanghai, China). Eluted proteins were examined by western blotting.

### 4.7. Mitochondrial Depolarisation Assessment

According to the manufacturer’s protocol, pretreated A2780 cells were collected and suspended in 1 mL of complete medium containing 10 µg/mL JC-1 (Beyotime) for 30 min at 37 °C. Cells were analysed with an Accuri C6 flow cytometer (BD Biosciences, Franklin Lakes, NJ, USA).

### 4.8. mt-Keima-COX8 A2780 Cell Line to Assess the Level of Mitochondrial Autophagy

#### 4.8.1. Construction of mt-Keima-COX8 A2780 Cell Line

The mt-Keima-COX8 lentivirus samples were purchased from Public Protein/Plasmid Library (Nanjing, China).

A cell line stably expressing mt-Keima was obtained by lentivirus infection. A2780 cells were inoculated on a 6-well culture plate. The cell confluence was about 50% when plating, and the volume of each culture medium was 2 mL. The cell fusion rate was about 70% during virus infection. Accurately dilute the lentivirus stock solution with the culture solution (can be diluted with serum-free culture solution). Add the calculated virus solution to the target cells and controls, mix them and put them in a carbon dioxide incubator (37 °C, 5% CO_2_). Change the culture medium after 8–16 h of virus infection; 48 h after virus-infected cells, conduct infected cells selection with the appropriate concentration of Puromycin.

#### 4.8.2. Flow Cytometry

Keima fluorescence signal (green) in neutral environment was detected with 440 nm excitation light, and Keima fluorescence signal (red) in acid environment was detected with 586 nm excitation light. Millipore Guava EasyCyte Flow Cytometer (Alsace, France). Accuri C6 Flow Cytometer (BD Biosciences).

#### 4.8.3. Fluorescence Microscopy Evaluation

Keima fluorescence signal (green) in neutral environment was observed with 440 nm excitation light, and Keima fluorescence signal (red) in acid environment was observed with 586 nm excitation light; Echo Revolve Hybrid Microscope (San Diego, CA, USA).

### 4.9. RT-PCR to Detect Mitochondrial Copy Number

Total mtDNA was extracted from A2780 cells using TRIzol (Invitrogen) according to the manufacturer’s protocol. Briefly, cDNA was synthesised using 0.5 µg of total RNA and SuperScript pre-amplification system (Promega). Reverse transcribed products were used to amplify ND1 by qRT-PCR. The primer sequences used for qRT-PCR were 5′-CACCCAAGAACAGGGTTTGT-3′ (forward) and 5′-TGGCCATGGGATTGTTGTTAA-3′(reverse) for ND1, and 5′-TAGAGGGACAAGTGGCGTTC-3′ (forward) and 5′-CGCTGAGCCAGTCAGTGT-3′ (reverse) for 18Rrna. All assays were per-formed in triplicate.

### 4.10. Real-Time Cell Analysis (RTCA)

The RTCA S16 System (ACEA Biosciences, San Diego, CA, USA) was used to monitor cellular status. The Principles and methods have been reported in previous studies. Briefly, cells were seeded into a special 16-well electronic plate (16-E-Plate). Then, the plate was placed into the special station and connected to an electronic sensor analyser by electrical cables. Then the station was placed in the culturing CO_2_ incubator. The more cells there are on the electrodes, the larger the change in electrode impedance. A unitless parameter termed Cell Index is used to measure the relative change in electrical impedance to represent cell status. The accompanying software was used to carry out the dimensionless impedance-based Cell Index. The Cell Index value changes with time and reflects the number of cells inside the well.

### 4.11. Flow Cytometry Analysis

Cell death was analysed using Annexin-V FITC/PI (BD Biosciences) staining. Cells were seeded in 6-well plates and treated with drugs as indicated. The attached and detached cells were harvested and subjected to Annexin-V FITC/PI staining according to the manufacturer’s instructions. Samples were analysed using Millipore Guava EasyCyte Flow Cytometer (Alsace, France).

### 4.12. Immunofluorescence and Microscopy

Cells were fixed in 4% (*w*/*v*) paraformaldehyde (PFA)/PBS for 20 min and then permeabilised with 0.1% Triton X-100 for 15 min. After blocking with bovine serum albumin for 30 min, cells were incubated with primary antibody overnight at 4 °C. Cells were then incubated with FITC conjugated secondary antibodies (Proteintech) at room temperature for 1 h; Echo Revolve Hybrid Microscope (San Diego, CA, USA).

Use image j to analyse the quality of co-localisation.

### 4.13. Statistical Analysis

Results are expressed as mean values ± standard deviation (SD). ANOVA was performed to compare results among groups. *p* < 0.05 was considered statistically significant. All experiments were repeated three times. Statistical analysis was performed with GraphPad Prism 7 (La Jolla, CA, USA).

## Figures and Tables

**Figure 1 ijms-22-03983-f001:**
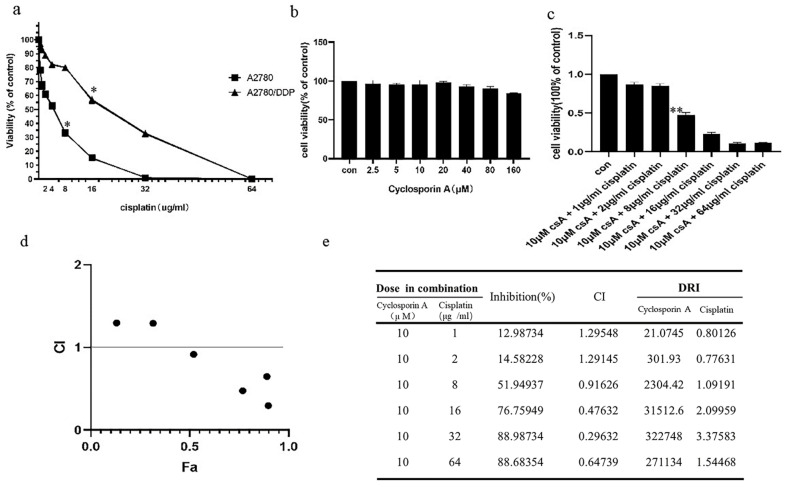
A2780 andA2780/DDP cells’ viability of cisplatin and the level of mitophage. (**a**). A2780 andA2780/DDP cells were treated with different doses of cisplatin. Cell viability was determined by MTT assay. Data are presented as mean ± SD, *n* = 3. ** p* < 0.05 vs. control. (**b**). A2780/DDP cells were treated with different doses of Cyclosporin A (CsA), Cell viability was determined by MTT assay. (**c**). A2780/DDP cells were treated with different doses of cisplatin and 10 μM Cyclosporin A (CsA) Cell viability was determined by MTT assay. Data are presented as mean ± SD, *n* = 3. *** p* < 0.05 vs. control. (**d**,**e**). Data in C were analysed by CompuSyn, CI: Combination index (mutually nonexclusive), CI < 1 indicates synergy, CI > 1 indicates antagonism, CI = 1 indicates additivity; Fa means the ratio of killed cells; DRI: Dose reduction index, Dose reduction index (DRI) means that when a certain level of inhibition is produced by the combination of drugs, the dose of a drug used is reduced by the multiple of the dose used when the drug is used alone to reach the same level of inhibition. (**f**). A2780 and A2780/DDP cells were treated with cisplatin (5 μg/mL) or FCCP (10 μM) for 8 h. Localisation of LC3II and mitotracker red was observed with fluorescence microscope. Scale bar, 10 µm. ①, ② and ③ are used to indicate the comparison between two different groups (e.g., ① A2780-con vs. A2780/DDP-con). Use image j to analyse pictures, Pearson correlation coefficient (POC) of A2780 was 0.3159631.POC of cisplatin treated A2780/DDP was 0.8165287. POC of FCCP-treated A2780/DDP was 0.8051136.

**Figure 2 ijms-22-03983-f002:**
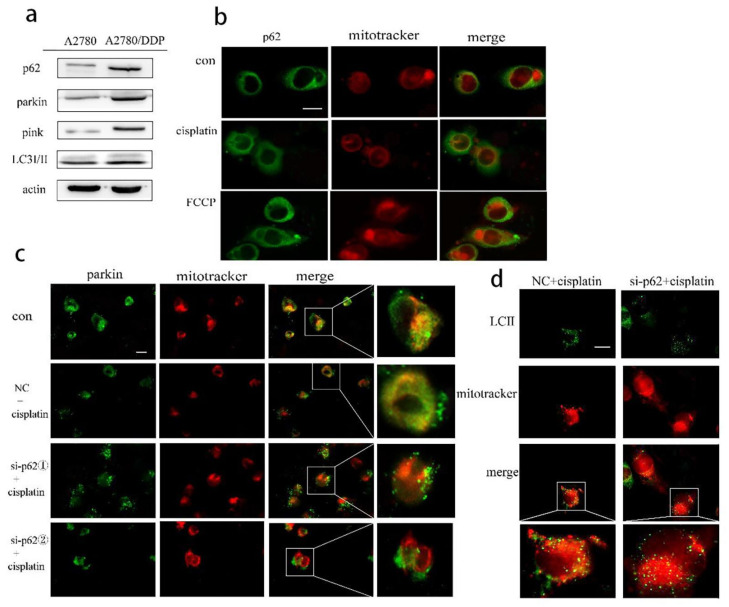
In ovarian cancer cells, p62 participates in parkin-mediated mitophagy stimulated by cisplatin. (**a**). Total proteins of A2780 and A2780/DDP cells were acquired, and the expression levels of p62, pink1, parkin and LC3 II/I were analysed by Western blot. (**b**). A2780/DDP cells were treated with cisplatin (20 μg/mL) for 8 h. Localisation of p62 and mitotracker red was observed with fluorescence microscope. Scale bar, 10 µm. (**c**). A2780/DDP cells transfected with si-p62. ① or si-p62. ② were treated with cisplatin (20 μg/mL) for 8 h. Localisation of parkin and mitotracker red was observed with fluorescence microscope. Scale bar, 10 µm. (**d**). A2780/DDP cells transfected with si-p62. ① were treated with cisplatin (20 μg/mL) for 8 h. Localisation of LC3II and mitotracker red was observed with fluorescence microscope. Scale bar, 10 µm.

**Figure 3 ijms-22-03983-f003:**
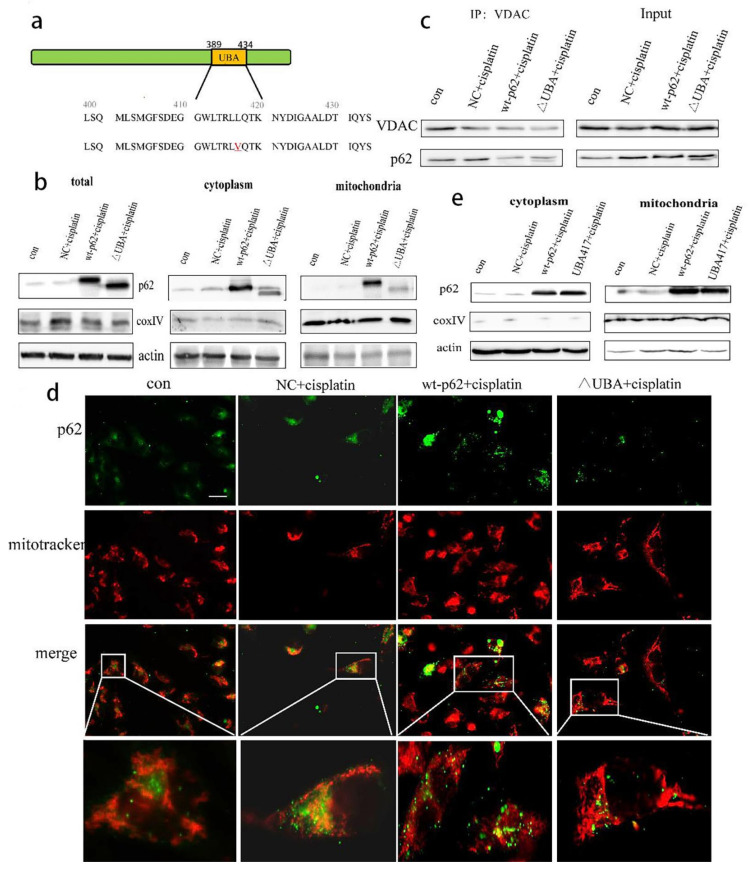
UBA domain determines the location of p62 on mitochondria. (**a**). Schematic representation of the p62 wild-type and mutant UBA domain constructs. (**b**). A2780 cells were transfected with wt-p62 or ΔUBA-p62. Western blot analysis of p62, actin and coxIV in cell lysate and mitochondrial lysate in A2780 after treatment (cisplatin 5 μg/mL, 6 h). (**c**) A2780 cells transfected with wt-p62 or ΔUBA-p62 were treated with cisplatin 5 μg/mL for 6 h. Immunoprecipitation was performed with the anti-VDAC1 antibody followed by Western blotting using anti-p62, anti-VDAC1. (**d**). A2780 cells were treated as in (**c**). Localisation of p62 and mitochondrial observed with microscopy. Scale bar, 10 μm. (**e**). A2780 cells transfected with wt-p62 or UBA417-p62 were treated with cisplatin 5 μg/mL for 6 h. Western blot analysis of p62, actin and coxIV in cell lysate and mitochondrial lysate.

**Figure 4 ijms-22-03983-f004:**
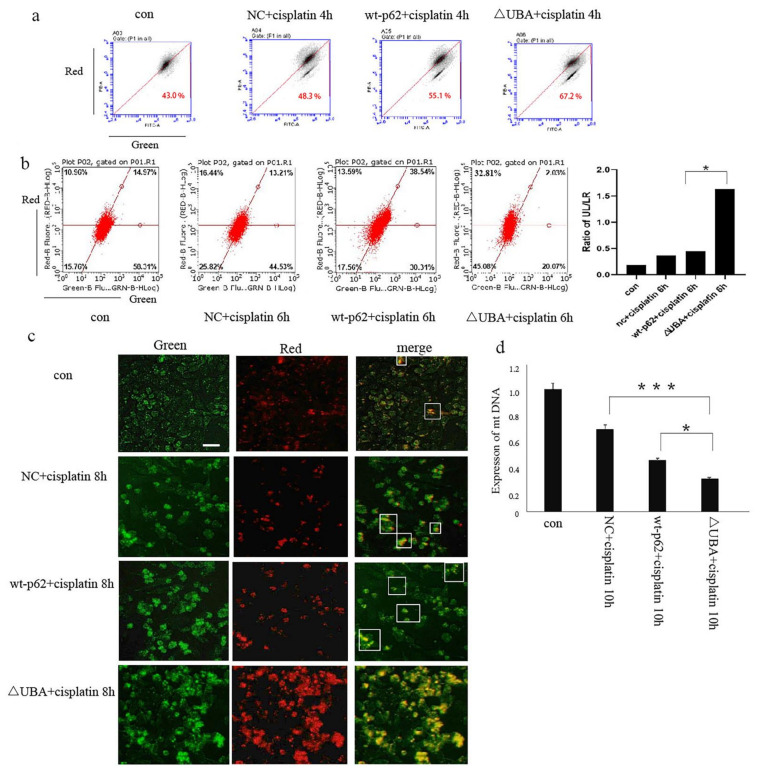
Mitophagy increases after UBA domain mutation. (**a**). JC-1was used to evaluate △φm in A2780 cells transfected with wt-p62 or ΔUBA-p62 and treated with cisplatin 5 μg/mL for 4 h. (**b**)., (**c**). After transducing lentivirus mt-keima-COX8, A2780 cells were transfected with wt-p62 or ΔUBA-p62, then treated with cisplatin 5 μg/mL for 6 h, fluorescence was visualised using microscopy and flow cytometry analysis. Scale bar, 50 μm. Data are presented as mean ± SD, *n* = 3. * *p* < 0.05 vs. wt-p62. (**d**). RT-PCR detection of mitochondrial copy number in A2780 cells transfected with wt-p62 or ΔUBA-p62 and treated with cisplatin 5 μg/mL for 10 h. Data are presented as mean ± SD, *n* = 3. ** p* < 0.05 vs. wt-p62. *** *p* < 0.001 vs. NC.

**Figure 5 ijms-22-03983-f005:**
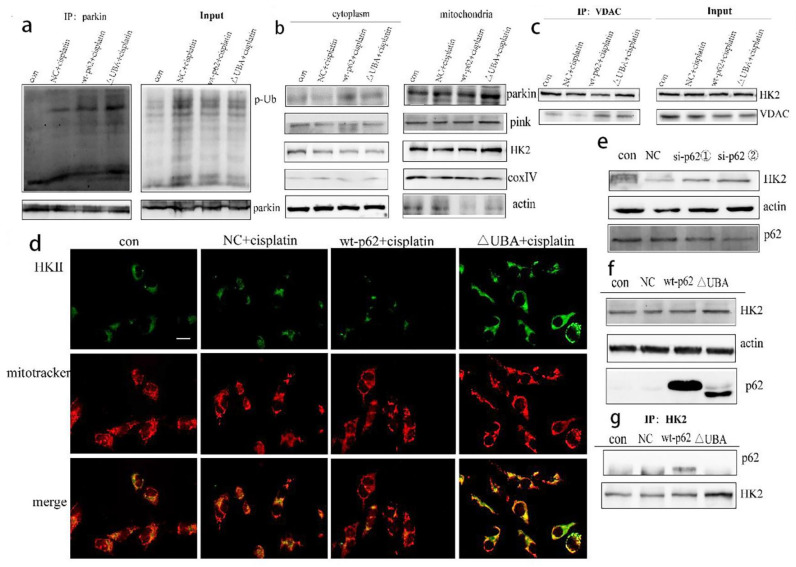
Increased recruitment of HK2 and p-Ub-parkin on mitochondria after mutating UBA domain. (**a**). A2780 cells transfected with wt-p62 or ΔUBA-p62 and treated with cisplatin 5 μg/mL for 8 h. Immunoprecipitation was performed with the anti-phosphorylated ubiquitin (p-Ub) antibody followed by Western blotting using anti-p-Ub, anti-HK2. (**b**). A2780 cells were transfected with wt-p62 or ΔUBA-p62. Western blot analysis of pink1, parkin, HK2, actin and coxIV in cell lysate and mitochondrial lysate in A2780 after treatment (cisplatin 5 μg/mL, 8 h). (**c**,**d**). A2780 cells transfected with wt-p62 or ΔUBA-p62 and treated with cisplatin 5 μg/mL for 6 h. The colocalisation between HK2 and mitochondrial was detected by immunoprecipitation and immunofluorescence, which was observed by Western blotting and microscope. Scale bar, 10 μm. (**e**,**f**). A2780 cells transfected with si-p62 and wt-p62 or ΔUBA-p62. The expression of HK2 was detected by Western blotting.si-p62①, ② were purchased from Sangon Biotech (**g**). A2780 cells transfected with wt-p62 or ΔUBA-p62 and treated with cisplatin 5 μg/mL for 12 h. Immunoprecipitation was performed with the anti-HK2 antibody followed by Western blotting using anti-p62, anti-HK2.

**Figure 6 ijms-22-03983-f006:**
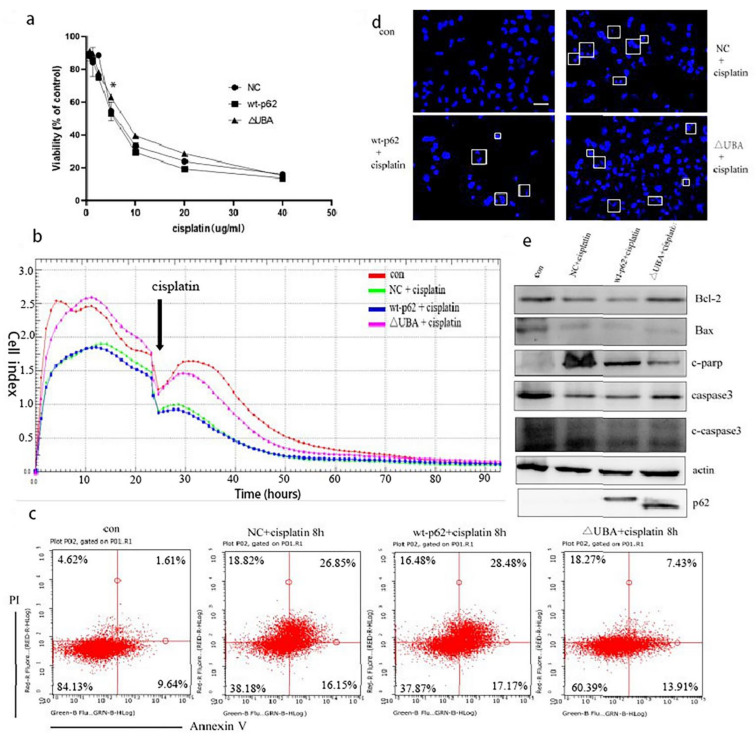
The sensitivity of A2780 cells to cisplatin decreased after UBA domain mutation. (**a**). A2780 cells transfected with wt-p62 or ΔUBA-p62 were treated with 5 μg/mL cisplatin for 24 h. Cell viability was determined by MTT assay. Data are presented as mean ± SD, *n* = 3. ** p* < 0.05 vs. wt-p62. (**b**). Time-dependent cell response profiles of cisplatin (5 μg/mL) treatment in ovarian cancer cells. The cells suspensions were transferred to E-Plates and placed on the RTCA reader for real-time monitoring every 5 min for the duration of the assay. The number of cells inside the well was displayed as Cell Index. The arrow indicates the point of cisplatin addition. (**c**). A2780 cells transfected with wt-p62 or ΔUBA-p62 were treated with 5 μg/mL cisplatin for 8 h, the apoptosis rate was detected by flow cytometry analysis. (**d**). A2780 cells transfected with wt-p62 or ΔUBA-p62 were treated with 5 μg/mL cisplatin for 8 h and stained with Hoechst 33342. Cell morphology was observed using confocal microscopy. indicate apoptotic cells. Scale bar, 50 μm. (**e**). Western blot analysis of c-PARP,c-caspase3,caspase3,Bcl2,Bax, expression after treatments as C.

**Figure 7 ijms-22-03983-f007:**
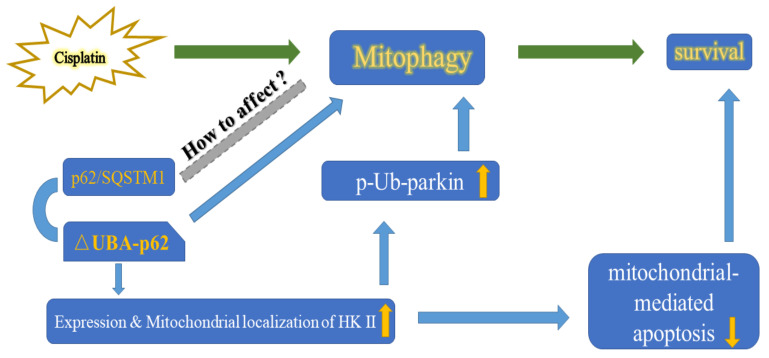
Proposed model by which the UBA domain deletion of p62 promotes the expression & mitochondrial localisition of HK2 then increases p-Ub-parkin during cisplatin induced mitophagy.

## Data Availability

Data sharing not applicable.

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
