# Peer review of "An Experimentally Induced Mutation in the UBA Domain of p62 Changes the Sensitivity of Cisplatin by Up-Regulating HK2 Localisation on the Mitochondria and Increasing Mitophagy in A2780 Ovarian Cancer Cells"

_ijms, 2021, doi:10.3390/ijms22083983_

Round 1

Reviewer 1 Report

The article entitled “A mutation in the UBA domain of p62 changes the sensitivity of cisplatin by up-regulating HK2 localisation on the mitochondria and increasing mitophagy in ovarian cancer cells” showed that a mutation in UBA domain of p62 sensitized apoptosis induced by cisplatin in ovarian cancer. The study in  line with the aims of the IJMS Journal.

The introduction provides sufficient background and includes relevant references. The methods are adequately described, the results are clearly described, and the discussion supported data.  The English language is appropriate.

So the paper is accepted in present form, without revision.

Author Response

Thank you very much for your review.

Reviewer 2 Report

In this manuscript, and using various molecular biology techniques in addition to IF single and double labeling methods, the authors investigated  the role of p62 in the enhanced prosurvival mitophagy by cisplatin in ovarian cancer cells. They found that mutations in UBA domain of the above mentioned protein upregulated HK2 in mitochondria resulting in increased phosphorylation of Parkin and  enhanced mitophagy. This work is interesting, but there are several issues to be answered by the authors:

1-Why did you use two ovarian cancer cell lines?

2-Please define  the mitophagosomes and mitophagolysosomes or mitolysosomes.I advise the authors to read and cite the guideline of autophagy research, 4th edition, which has been published in autophagy journal last month.

3-Figure 1 is poor, please resubmit better figure showing clearly the diagrams from a to e.

4-What is the subcellular localization  under normal and cisplatin treatment of HK2? If mitochondrial, at which membrane it is accumulated? 

5-It is better to submit additional diagram showing a schema summarizing the results and conclusions

6-In materials and methods, in IF and Western blot  sections, please add the names and catalog numbers of antibodies used and the producers.

7-Did you try to use TEM or immunoelectron microscopy to detect mitophagosomes  labelled with Parkin?

8-Do you have any in vivo evidence supporting your data?

9-How cisplatin could induce  mutations of the UBA domain of p62? 

Reviewer 3 Report

In this manuscript, the authors seek to study the cisplatin resistance mechanism in ovarian cancer cells. They focused on the mechanism of autophagy. These types of studies are interesting and needed to improve the effectiveness of chemotherapy. However, this manuscript contains too many errors and ambiguities in results interpretation. I would like to point out some of them:
1. Many conclusions rely on western blot analysis of proteins level. In those cases, authors should conduct densitometric analysis.
2. Statistical analysis in many cases of cytometric analysis, cytotoxicity assay, cell index tests were not conducted. However, in some cases, the authors compared results treatment type after those experiments.
3. Compusyn software is used to quantification the value of the combination index (not to analyze of cell index). The two drugs always exert a combined effect independent of CI value. CI value is calculated dependent on the combined effect value of those drugs.
4. The results of cytometry analysis are presented in low-quality cytograms. The numeric values of results are invisible. The results are unclearly described. 

5. The authors do not interpret or incorrectly interpret many results. E.g: "We used annexin V/propidium iodide staining and western blotting to detect apoptosis in each group after adding cisplatin and observed a proportion of apoptotic cells in theΔUBA group (Fig. 6.d). At the same time, the ratio of Bcl2 to Bax increased and the expression of cleaved caspase 3 (c-caspase 3) and cleaved poly (ADP-ribose) polymerase (c-PARP) decreased (Fig. 6.e).6. The aim of the studies was not composed. 
7. In the discussion section, the authors not fully discussed the results of the experiments. E.g. they only mentioned apoptosis experiments. They did not interpret those. 
8. The usage of A2780 and A2780/DDP cell lines did not justify by authors in the manuscript text. 

Round 2

Reviewer 2 Report

The manuscript is greatly improved

I think it is acceptable.

Reviewer 3 Report

The manuscript can be accepted for publication in the present form.